# Exploring Flood-Related Unintentional Fatal Drowning of Children and Adolescents Aged 0–19 Years in Australia

**Amy E. Peden** [1,2,3,*] and **Richard C. Franklin** [1,2]

1   Royal Life Saving Society–Australia, Broadway, NSW 2007, Australia
2   College of Public Health, Medical and Veterinary Sciences, James Cook University, Townsville, QLD 4811, Australia
3   School of Public Health and Community Medicine, University of New South Wales, Sydney, NSW 2033, Australia
*   Correspondence: apeden@rlssa.org.au

**Abstract:** Disasters, such as flooding, are predicted to increase. Drowning is one of the leading causes of death during times of flood. This study examined the little explored topic of child drowning during floods, with the aim of identifying risk factors to inform prevention strategies. A retrospective, total population examination of cases of children and adolescents aged 0–19 years who died from unintentional flood-related drowning in Australia for the 16-year period 1 July 2002 to 30 June 2018 was undertaken. Univariate and chi-square analysis was conducted, with Fisher's exact test used for cell counts <5. Across the study period, 44 flood-related drowning deaths occurred among children and adolescents (63.6% male; 34.1% aged 10–14 years). Almost all (84.1%) occurred in rivers, creeks, or streams in flood, with the remaining incidents occurring in storm water drains (n = 7). Leading activities immediately prior to drowning were non-aquatic transport (40.9%), swimming in floodwaters (25.0%), and falls into floodwaters (15.9%). Flood-related fatal drowning among children and adolescents is rare (0.05 per 100,000 population), however flood-drowning risk increases as remoteness increases, with children and adolescents drowning in floodwaters in very remote areas at a rate 57 times that of major cities. All drownings are preventable, and this study has identified key causal factors that must be considered in advocacy and prevention efforts. These include: the importance of adult supervision, avoiding flooded waterways when driving or for recreational purposes, and the increased risks for those residing in geographically isolated and socially disadvantaged areas. Findings must be considered when developing interventions and advocacy for the purposes of the reduction of child and adolescent drowning during times of flood.

**Keywords:** drowning; child; disaster; flooding; epidemiology; injury prevention; risk reduction; resilience; adolescent

---

## 1. Introduction

Children and young people are at increased risk of drowning, especially during times of flood [1]. The World Health Organization (WHO) reports that, globally, over half of all drowning deaths are among those aged under 25 years [2]. In particular, children aged under five (0–4 years) are disproportionately at risk [2], most commonly drowning in swimming pools [3,4] and bathtubs [5] in high income contexts, and in open bodies of water around the home in low income contexts [6].

The WHO estimates the global drowning burden to be 360,000 deaths per year [7]. However, this estimate of unintentional fatal drowning, using International Classification of Diseases (ICD) codes W65–74 only, excludes flood-related drowning fatalities (ICD codes X37—cataclysmic storm and X38—exposure to flooding) [8]. Flooding is the most destructive of all natural disasters with respect to

the number of people affected and the resultant economic losses [9]. In 2018, floods resulted in the deaths of 2859 people and affected a further 35.4 million people [10].

Drowning is one of the leading causes of death during times of flood [11], with mortality and morbidity likely to grow with extreme weather and floods increasing due to the effects of climate change [12]. In 2018, flooding resulted in the deaths of 504 people in India, 220 in Japan, 199 in Nigeria and 151 in the Republic of Korea [10]. Examination of flood-related drowning deaths between 1980 and 2009 identified 539,811 deaths with motor vehicle incidents and male gender associated with increased mortality in developed countries, and female gender likely linked to higher mortality in low-income countries [12].

In Australia, there are an average of 13 flood-related unintentional drowning deaths annually, with an annual average of three flood-related drowning deaths among children and adolescents 0–19 years of age [13]. Slow onset flooding is most common, accounting for over half (56%) of all drowning fatalities; flash flooding is responsible for a further 27% of deaths [13]. Males are overrepresented in statistics; however, females are at higher risk of flood-related drowning than non-flood related drowning [13]. Driving into floodwaters, being swept away by flash flooding and falls into floodwaters are the most common activities being undertaken immediately prior to flood-related drowning in Australia [13].

While previous bodies of research associated with floods have focused on the overall epidemiology [13–15], behavioral research associated with driving into floodwaters [16–20], and issues impacting rescuers [21]—as well as community experiences [22], perceptions [23], response and resilience [24,25]—few studies have specifically examined the epidemiology and circumstances associated with child drowning during times of flood. A study in Bangladesh identified a strong association between the annual monsoon season and child drowning of those aged 1–4 years in Matlab, Bangladesh [26]. Data from Nepal identifies children as recording flood-related fatality rates six times higher than mortality rates in the same village prior to flood [27]. Data from the United States suggests young adults (aged 10–19) have a higher vulnerability to flooding [1].

To address this knowledge gap from an Australian perspective, this study aims to explore the epidemiology of unintentional fatal child and adolescent drowning during times of flood, with a particular focus on causal factors to inform preventative efforts.

## 2. Materials and Methods

All unintentional, flood-related drowning deaths in Australia of children and adolescents aged 0–19 years [28] were extracted from the Royal Life Saving National (Australian) Fatal Drowning Database (the Database), which uses a data triangulation method of case capture. This process has been documented previously [8]. However, in brief, year-round media monitoring is conducted to identify cases of suspected drowning within the media. These data are cross-referenced with data from child death review teams and lifesaving organizations and the National Coronial Information System (NCIS) [29]. Details on the circumstances and causal factors of the flood-related drowning are drawn from a police report and/or coronial finding. Data analyzed covers a 16-financial-year period (2002/03 to 2017/18), the maximum number of years of data currently available in the Database. In Australia, financial years run from 1 July to 30 June.

In Australia, all sudden and unexpected deaths (such as drowning) must be reported to and investigated by a coroner. While under investigation, a coroner will consider police, autopsy, and toxicology reports to determine the circumstances and cause of death. Once determined, a coroner's report is completed, and the case is closed on the NCIS. Cases may go to coronial inquest where recommendations are made to prevent future loss of life in similar circumstances [30].

Blood alcohol concentrations (BAC) were examined using toxicological reports. A positive BAC was defined as any detection of blood alcohol (i.e., a BAC ≥ 0.001%); and a contributory blood alcohol level was defined as a BAC ≥ 0.050% [31].

This study used definitions adopted by Jonkman and Kelman (2005) [32] namely, a flood is defined as the presence of water in areas that are usually dry and a flood fatality (in this study one that occurred due to drowning) is defined as a fatality that would not have occurred without a specific flood event. Cases of flood-related drowning were identified through the use of keywords and ICD coding. Cases where the word 'flood' was mentioned in the police report or coroners' report to describe the circumstances of the incident were coded as flood-related – yes. Similarly, any cases during the study period which had an ICD code of X38—flood and X37—cataclysmic storm were included in the dataset for analysis [33]. The activity of non-aquatic transport relates to vehicles not intended to be in the water i.e., cars, trucks, machinery, bicycles [34]. In this instance, vehicles were either intentionally driven through floodwaters or swept into floodwaters in the case of flash flooding.

The impact of social determinants of health on child flood drowning risk were examined through the use of remoteness classification of drowning incident location and the Index of Relative Socio-economic Advantage and Disadvantage (IRSAD) [35] of the child's residential postcode. IRSAD is an index which summarizes the economic and social conditions of people and households within an area, including both relative advantage and disadvantage measures. The index is ranked from 1 to 10, with a low score indicating relatively greater disadvantage (e.g., many people with low incomes and many people in unskilled occupations), compared to a high score which indicates a relative lack of disadvantage [35]. For analysis, IRSAD was categorized as low (rank 1–3), mid (rank 4–7) and high (rank 8–10).

The remoteness classification of drowning incident was coded to the five Australian Standard Geographical Classifications (ASGC): Major cities, inner regional, outer regional, remote, and very remote [36]. Seasons in Australia are Summer (December to February); Autumn (March to May); Winter (June to August); and Spring (September to November). Time of day of drowning incident is coded into the following categories: Early Morning (12:00 a.m. to 6:00 a.m.); Morning (6:01 a.m. to 12:00 p.m.); Afternoon (12:01 p.m. to 6:00 p.m.); and Evening (6:01 p.m. to 12:00 a.m.).

Multiple fatality events (MFEs) are defined as such where more than one person drowns in the same incident. All Terrain Vehicles (ATVs), also known as quad bikes, are generally three- or four-wheel vehicles that travel on low-pressure tires, with a seat that is straddled by the operator, along with handlebars for steering control. ATVs are used extensively in agriculture and for trail riding for recreation [37]. A 4WD (also known as a 4 × 4) is a two-axled vehicle capable of providing torque to all of its wheels simultaneously.

Population data on children and adolescents aged 0–19 years in Australia were sourced from the Australian Bureau of Statistics (ABS) [38]. Population data by remoteness classification is only available in datasets generated by the Australian population census. Therefore, a two yearly average was calculated using data from the four census years (2011 and 2016) [39,40] and used with a 16 yearly average of the deaths to calculate crude annualized fatal drowning rates for children and adolescents during times of flood per 100,000 population.

*Statistical Analyses and Ethics*

Univariate and chi-square analyses were undertaken, with statistical significance deemed $p < 0.05$. For cells with small counts (i.e., <5) in Table 1, a Fisher's exact test was used. Non-parametric analysis was conducted using the proportional basis of the population as the assumed outcome numbers. Relative risk (RR) was calculated with a 95% confidence interval (CI) for rates of drowning for each remoteness classification, with the classification with the lowest rate used as the control. Due to ethical constraints around reporting small numbers, cells with counts <4 are concealed with NP (Not Presented).

Ethics approval for this study was given by the Victorian Department of Justice and Regulation Human Research Ethics Committee (CF/07/13729, CF/10/25057, CF/13/19798).

## 3. Results

Across the 16-year study period, a total of 44 children and adolescents aged 0–19 years drowned due to flooding; this represents a crude drowning rate of 0.05 per 100,000 population per annum (Figure 1). Males accounted for 63.6% of the child and adolescent flood-related drowning cases, although males were not found to be statistically overrepresented (Table 1).

**Table 1.** Demographics and circumstances of child and adolescent unintentional flood-related drowning deaths by sex of drowning victim, Australia, 2002/03–2017/18 (N = 44).

| | Total | | Male | | Female | | X2 (*p* Value) |
|---|---|---|---|---|---|---|---|
| | **N** | **%** | **N** | **%** | **N** | **%** | |
| Total | 44 | 100.0 | 28 | 63.6 | 16 | 36.4 | 2.680 (*p* = 0.102) |
| *Age group* | | | | | | | |
| 0–4 years | 7 | 15.9 | NP | 42.9 | 4 | 57.1 | 0.236 (*p* = 0.205) * |
| 5–9 years | 12 | 27.3 | 7 | 58.3 | 5 | 41.7 | 0.732 (*p* = 0.456) * |
| 10–14 years | 15 | 34.1 | 10 | 66.7 | 5 | 33.3 | 1.000 (*p*= 0.516) * |
| 15–17 years | 10 | 22.7 | 8 | 80.0 | NP | 20.0 | 0.283 (*p* = 0.200) * |
| *Location of drowning incident* | | | | | | | |
| River/Creek/Stream | 37 | 84.1 | 21 | 56.8 | 16 | 43.2 | *0.037 (p = 0.031) ** |
| Drain | 7 | 15.9 | 7 | 100.0 | 0 | 0.0 | |
| *Type of flooding* | | | | | | | |
| Flash flood | 14 | 31.8 | 6 | 42.9 | 8 | 57.1 | *7.418 (p = 0.013) ** |
| Slow onset | 13 | 29.5 | 12 | 92.3 | NP | 7.7 | |
| Unknown | 17 | 38.6 | 10 | 58.8 | 7 | 41.2 | - |
| *Activity undertaken immediately prior to drowning* | | | | | | | |
| Fall | 7 | 15.9 | 5 | 71.4 | NP | 28.6 | 1.000 (*p* = 0.496) * |
| Jumped In | NP | 4.5 | NP | 100.0 | 0 | 0.0 | 0.526 (*p* = 0.400) * |
| Non-aquatic Transport | 18 | 40.9 | 10 | 55.6 | 8 | 44.4 | 0.525 (*p* = 0.466) * |
| Swept Away | 3 | 6.8 | NP | 33.3 | NP | 66.7 | 0.543 (*p* = 0.296) * |
| Swimming and Recreating | 11 | 25.0 | 7 | 100.0 | 0 | 0.0 | 1.000 (*p* = 0.635) * |
| Watercraft | NP | 6.8 | NP | 100.0 | 0 | 0.0 | 0.290 (*p* = 0.247) * |
| *Season of drowning incident* | | | | | | | |
| Summer | 24 | 54.5 | 17 | 70.8 | 7 | 29.2 | 0.352 (*p* = 0.220) * |
| Autumn | 7 | 15.9 | 4 | 57.1 | NP | 42.9 | 0.692 (*p*= 0.504) * |
| Winter | 5 | 11.4 | NP | 60.0 | NP | 40.0 | 1.000 (*p* = 0.608) * |
| Spring | 8 | 18.2 | 4 | 50.0 | 4 | 50.0 | 0.434 (*p* = 0.310) * |
| *Time of day of drowning incident (n = 42)* | | | | | | | |
| Early Morning | 0 | 0.0 | 0 | 0.0 | 0 | 0.0 | UTBC |
| Morning | 8 | 19.0 | 6 | 75.0 | NP | 25.0 | 0.697 (*p* = 0.457) * |
| Afternoon | 26 | 61.9 | 18 | 69.2 | 8 | 30.8 | 0.742 (*p* = 0.452) * |
| Evening | 8 | 19.0 | 4 | 50.0 | 4 | 50.0 | 0.406 (*p* = 0.240) * |
| *Remoteness classification of incident location* | | | | | | | |
| Major Cities | 9 | 20.5 | 7 | 77.8 | NP | 22.2 | 0.450 (*p* = 0.280) * |
| Inner Regional | 13 | 29.5 | 8 | 61.5 | 5 | 38.5 | 1.000 (*p* = 0.557) * |
| Outer Regional | 11 | 25.0 | 8 | 72.7 | NP | 27.3 | 0.719 (*p* = 0.365) * |
| Remote | 4 | 9.1 | NP | 75.0 | NP | 25.0 | 1.000 (*p* =0.537) * |
| Very Remote | 7 | 15.9 | NP | 28.6 | 5 | 71.4 | *0.080 (p = 0.049) ** |
| *IRSAD classification of child's residential location* | | | | | | | |
| Low | 19 | 43.2 | 11 | 57.9 | 8 | 42.1 | 0.540 (*p* = 0.353) * |
| Mid | 23 | 52.3 | 15 | 65.2 | 8 | 34.8 | 1.000 (*p* = 0.533) * |
| High | NP | 4.5 | NP | 100.0 | NP | NP | 0.526 (*p* = 0.400) * |

* Denotes where Fishers exact test has been used. UTBC = Unable to be calculated. NP = Not presented.

When examining flood-related fatal drowning among children and adolescents by age group, the largest proportion of drowning deaths involved those aged 10–14 years (34.1%). When examining drowning rates, the 10–14 years age group also recorded the highest rate at 0.07 per 100,000 population, compared to 0.03 per 100,000 population among those aged 0–4 years (Figure 1).

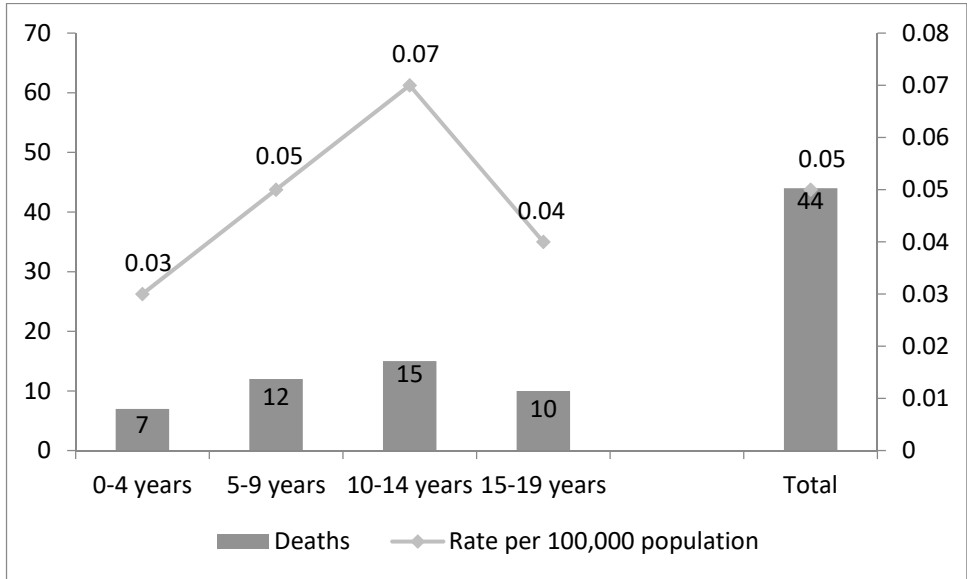

**Figure 1.** Crude annualized incidence and rate of child and adolescent flood-related fatal unintentional drowning by age group, Australia, 2002/03–2017/18 (N = 44).

More females than males were involved in flood-related drowning among the 0–4 years age group (57.1% female), while in all other age groups, males were represented more highly than females, with double the number of male drownings when compared to females in the 10–14 years age group (66.7% male) and four times the number of drowning deaths among males in the 15–19 years age group (80.0% male) (Table 1).

Rivers, creeks, and streams were the leading category of aquatic location for flood-related drowning (84.1%) with the remaining incidents occurring in drains. Females were significantly more likely to drown in rivers while males were significantly more likely to drown in drains ($X^2$ = 0.04; $p$ = 0.031). Females were significantly more likely to be involved in flash flood-related drowning incidents ($X^2$ = 7.42; $p$ = 0.013) (Table 1).

The largest proportion of flood-related river drowning deaths occurred among children aged 10–14 years (35.1%). Young children (0–4 years) commonly drowned as a result of a fall into a flooded river (42.9%), while non-aquatic transport incidents accounted for the highest proportion of flood-related drowning incidents among children 5–9 years (70.0%) and 10–14 years (53.8%). Younger children (0–9 years) were more likely to be involved in flash flooding incidents ($X^2$ = 8.98; $p$ = 0.030). Drowning as a result of swimming and recreating in flooded rivers was most common among 10–14-year-olds (n = 5; 38.5% of all deaths in this age group) and 15–19-year-olds (n = 2; 28.6% of all drowning deaths in this age group) (Figure 2).

When examining the seven drowning incidents in drains, common themes emerged. All (100.0%) drain drowning fatalities involved males (Table 1). Seventy one percent (71.4%) of incidents occurred in those aged 10–19 years (Table 2). No young child (0–4 years) drowned in a drain during times of flood (Table 2). Among younger children (5–9 years) falls or jumping into flooded drains were common. Among older children and adolescents (10–19 years) swimming or recreating in floodwaters was most common, prior to being sucked into a drain by the force of the water and drowning.

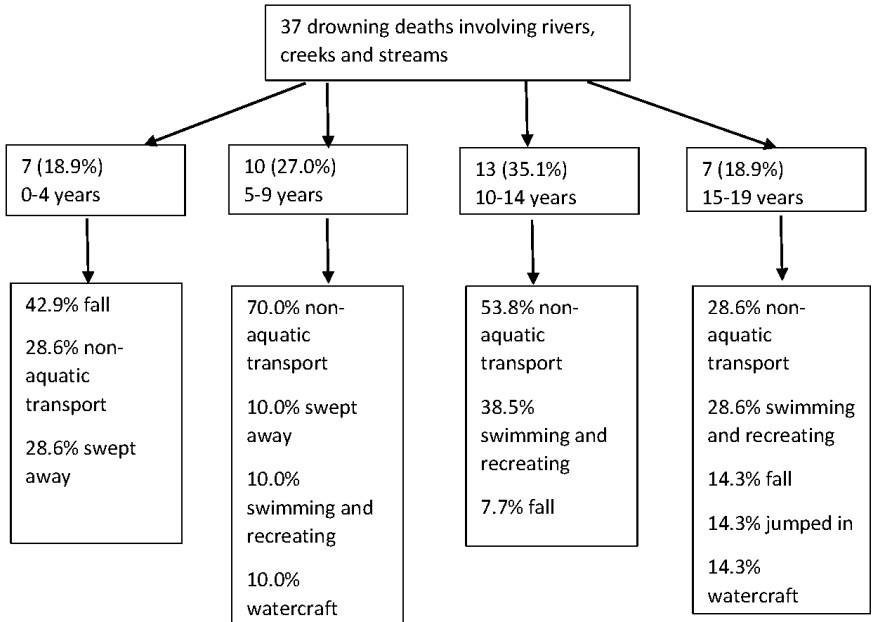

**Figure 2.** Flow chart exploring river flood related child and adolescent drowning deaths, activities presented in rank order, 0–19 years by age group, Australia, 2002/03–2017/18 (n = 37).

**Table 2.** Demographics and circumstances of child and adolescent unintentional flood-related drowning deaths, by age group of drowning victim, Australia, 2002/03–2017/18 (N = 44).

| | Total | | 0–4 Years | | 5–9 Years | | 10–14 Years | | 15–19 Years | | $X^2$ (*p* Value) |
|---|---|---|---|---|---|---|---|---|---|---|---|
| | N | % | N | % | N | % | N | % | N | % | |
| Total | 44 | 100.0 | 7 | 15.9 | 12 | 27.3 | 15 | 34.1 | 10 | 22.7 | 3.333 (*p* = 0.343) |
| *Sex* | | | | | | | | | | | |
| Male | 28 | 63.6 | NP | 10.7 | 7 | 25.0 | 10 | 35.7 | 8 | 28.6 | 2.669 (*p* = 0.446) |
| Female | 16 | 36.4 | 4 | 25.0 | 5 | 31.3 | 5 | 31.3 | NP | 12.5 | |
| *Location of drowning incident* | | | | | | | | | | | |
| River/Creek/Stream | 37 | 84.1 | 7 | 18.9 | 10 | 27.0 | 13 | 35.1 | 7 | 18.9 | 2.888 (*p*= 0.409) |
| Drain | 7 | 15.9 | 0 | 0.0 | NP | 28.6 | NP | 28.6 | NP | 42.9 | |
| *Type of flooding* | | | | | | | | | | | |
| Flash flooding | 14 | 31.8 | 5 | 35.7 | 5 | 35.7 | NP | 21.4 | NP | 7.1 | *8.975 (p = 0.030)* |
| Slow onset | 13 | 29.5 | NP | 7.7 | NP | 7.7 | 6 | 46.2 | 5 | 38.5 | |
| Unknown | 17 | 38.6 | NP | 5.9 | 6 | 35.3 | 6 | 35.3 | 4 | 23.5 | - |
| *Activity undertaken immediately prior to drowning* | | | | | | | | | | | |
| Fall | 7 | 15.9 | NP | 42.9 | NP | 14.3 | NP | 14.3 | NP | 28.6 | 5.397 (*p* = 0.145) |
| Jumped In | 2 | 4.5 | 0 | 0.0 | NP | 50.0 | 0 | 0.0 | NP | 50.0 | 2.130 (*p* = 0.546) |
| Non-aquatic Transport | 18 | 40.9 | NP | 11.1 | 7 | 38.9 | 7 | 38.9 | NP | 11.1 | 3.962 (*p* = 0.266) |
| Swept Away | 3 | 6.8 | NP | 66.7 | NP | 33.3 | 0 | 0.0 | 0 | 0.0 | 7.086 (*p*=0.069) |
| Swimming and Recreating | 11 | 25.0 | 0 | 0.0 | NP | 9.1 | 6 | 54.5 | 4 | 36.4 | 7.111 (*p* = 0.068) |
| Watercraft | NP | 6.8 | 0 | 0.0 | NP | 33.3 | NP | 33.3 | NP | 33.3 | 0.715 (*p* = 0.870) |
| *Season of drowning incident* | | | | | | | | | | | |
| Summer | 24 | 54.5 | NP | 12.5 | 6 | 25.0 | 8 | 33.3 | 7 | 29.2 | 1.458 (*p* = 0.692) |
| Autumn | 7 | 15.9 | 0 | 0.0 | NP | 28.6 | 4 | 57.1 | NP | 14.3 | 2.888 (*p* = 0.409) |
| Winter | 5 | 11.4 | NP | 40.0 | NP | 20.0 | NP | 40.0 | 0 | 0.0 | 3.507 (*p* = 0.320) |
| Spring | 8 | 18.2 | NP | 25.0 | NP | 37.5 | NP | 12.5 | NP | 25.0 | 2.242 (*p* = 0.524) |
| *Time of day of drowning incident (n = 42)* | | | | | | | | | | | |
| Early Morning | 0 | 0.0 | 0 | 0.0 | 0 | 0.0 | 0 | 0.0 | 0 | 0.0 | UTBC |
| Morning | 8 | 19.0 | NP | 25.0 | NP | 25.0 | 4 | 50.0 | 0 | 0.0 | 4.015 (*p* = 0.260) |
| Afternoon | 26 | 61.9 | 4 | 15.4 | 7 | 26.9 | 8 | 30.8 | 7 | 26.9 | 0.535 (*p* = 0.911) |
| Evening | 8 | 19.0 | 0 | 0.0 | NP | 37.5 | NP | 25.0 | NP | 37.5 | 2.671 (*p* = 0.445) |

**Table 2.** *Cont.*

| | Total | | 0–4 Years | | 5–9 Years | | 10–14 Years | | 15–19 Years | | X² (*p* Value) |
|---|---|---|---|---|---|---|---|---|---|---|---|
| | N | % | N | % | N | % | N | % | N | % | |
| Remoteness classification of incident location | | | | | | | | | | | |
| Major Cities | 9 | 20.5 | NP | 11.1 | NP | 33.3 | NP | 33.3 | NP | 22.2 | 0.319 (*p* = 0.956) |
| Inner Regional | 13 | 29.5 | 4 | 30.8 | NP | 15.4 | NP | 23.1 | 4 | 30.8 | 4.699 (*p* = 0.195) |
| Outer Regional | 11 | 25.0 | 0 | 0.0 | 5 | 45.5 | NP | 27.3 | NP | 27.3 | 4.444 (*p* = 0.217) |
| Remote | 4 | 9.1 | NP | 50.0 | 0 | 0.0 | NP | 50.0 | 0 | 0.0 | 5.741 (*p* = 0.125) |
| Very Remote | 7 | 15.9 | 0 | 0.0 | NP | 28.6 | 4 | 57.1 | NP | 14.3 | 2.888 (*p* = 0.409) |
| IRSAD classification of child's residential location | | | | | | | | | | | |
| Low | 19 | 43.2 | 5 | 26.3 | 3 | 15.8 | 6 | 31.6 | 5 | 26.3 | 4.145 (*p* = 0.246) |
| Mid | 23 | 52.3 | 2 | 8.7 | 8 | 34.8 | 8 | 34.8 | 5 | 21.7 | 2.600 (*p* = 0.457) |
| High | NP | 4.5 | 0 | 0.0 | NP | 50.0 | NP | 50.0 | 0 | 0.0 | 1.362 (*p* =0.714) |

NP = Not Presented. UTBC = Unable To Be Calculated.

Non-aquatic transport incidents (40.9%) and swimming and recreating in floodwaters (25.0%) were common activities preceding flood-related drowning. Young children (0–4 years) commonly accidentally fell into floodwaters (42.9% of all 0–4 years flood-related drowning fatalities), as opposed to those aged 15–19 years who were commonly swimming and recreating (40.0%) in floodwaters immediately prior to drowning (Table 2). In all instances, the adolescents swimming in floodwaters, did so for recreational or risk-taking purposes, and none were swimming for the purposes of evacuation.

Of the 18 non-aquatic transport incidents, 55.6% of incidents occurred on roads that were open at the time of the incident, with a further 11.1% of incidents occurring on a private road (i.e., private trail). No children (0.0%) were driving a vehicle when they drowned. In over half of incidents (55.6%) the driver also drowned. In 44.4% of incidents, the vehicle involved was a passenger car. In a further 27.8% cases, the vehicle was a 4WD (four wheel drive). Trucks, all-terrain vehicles (ATVs), AND motorcycles were the vehicle types involved in the remaining incidents. In nine instances (50.0%) the vehicles were involved in flash flood incidents. In 10 cases (55.6%), the vehicle was swept into floodwaters and in the remaining 44.4% of cases, the vehicle was intentionally driven into floodwaters. A contributory level of alcohol and illicit drugs were known to have affected the driver of the vehicle in 27.8% of deaths (Figure 3). Fifteen children and adolescents drowned in MFEs across 7 incidents.

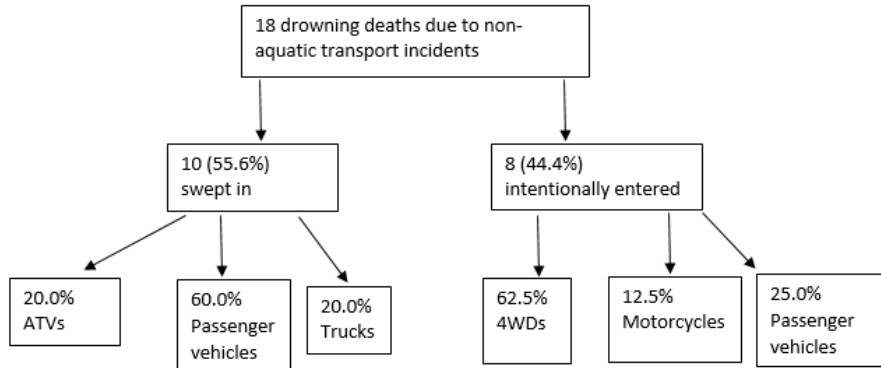

**Figure 3.** Flow chart exploring non-aquatic transport flood related child and adolescent drowning deaths, 0–19 years by age group, Australia, 2002/03–2017/18 (n = 18).

Summer (45.0%), followed by Spring (18.2%) were the most common seasons for flood-related drowning (Table 2). Over three quarters (79.5%) of drowning deaths occurred during the months of November to April inclusive (the traditional wet season in northern parts of Australia).

Flood-related child and adolescent drowning most commonly occurred in the afternoon (n = 26; 61.9%), followed by the morning (n = 8; 19.0%) (Table 1). Females were significantly more likely to drown in very remote areas, when compared to males (X² = 0.08; *p* = 0.049). The highest rates

of flood-related child and adolescent drowning were recorded in very remote areas (1.21 per 100,000 population). This represents a relative risk that is 57 times higher than in major cities (RR = 56.71; CI: 21.12–152.27) (Table 3). The largest proportion of flood-related drowning deaths among children and adolescents occurred in areas classified as mid (n = 23; 52.3%) and low socio-economic (IRSAD) (n = 19; 43.2%) (Table 2).

**Table 3.** Incidents and rates of child and adolescent flood-related drowning per 100,000 population by remoteness classification, Australia, 2002/03–2017/18 (N = 44).

|  | **Number of Drowning Deaths** | **Rate/100,000 Population** | **Relative Risk (RR) (95% Confidence Interval)** |
|---|---|---|---|
| Major Cities | 9 | 0.02 | 1 |
| Inner Regional | 13 | 0.11 | 5.38 (2.30–12.58) |
| Outer Regional | 11 | 0.20 | 9.44 (3.91–22.79) |
| Remote | 4 | 0.48 | 22.52 (6.94–73.13) |
| Very Remote | 7 | 1.21 | 56.71 (21.12–152.27) |

## 4. Discussion

Children are often the first victims of natural disasters [41], yet there has been little epidemiological research examining the circumstances and causal factors leading to flood-related drowning among this cohort. This study examined 44 cases of fatal unintentional child and adolescent drowning in floodwaters from Australia across a 16-year period. While this number is small, these data identified a range of causal factors impacting risk; in particular, two areas where drowning could have been prevented; i.e., driving or recreating in floodwaters or playing in drains. This study also found a significant increase in the relative risk of people drowning in floods as remoteness increases. Key findings are now discussed within the context of prevention.

### 4.1. Flood-Related Child Drowning Risk Profile by Age

Internationally, children have been identified as being at high risk of flood-related drowning [1,26,27,41]. This study has identified flood-related drowning risk differs by age, with peak risk identified among 10–14 year olds, demonstrating a need for age specific prevention strategies.

Flood-related drowning among young children often results from unintentional entry into floodwaters, with falls into water accounting for 50.0% of all flood-related drowning deaths among children 0–4 years. Such incidents highlight the importance of active adult supervision at all times. Active adult supervision is a vital part of a multi-faceted drowning prevention strategy for young children [5,42]; however, other strategies will be required, particularly for older children.

For natural bodies of water, fencing (restricting access) of the water body is often not practical. As such, a child safe play area (i.e., a barrier around the child) is a better strategy [42]. For flooding, which, by its definition, involves water in areas where it is not normally found [32], restricting access should supervision lapse, is even more challenging.

### 4.2. Risks of Driving into Floodwaters

Activity prior to drowning in floodwaters among children and adolescents is poorly understood. This study identified non-aquatic transport incidents (driving into floodwater) as the leading activity accounting for 41% of all deaths. Multiple prevention strategies have been suggested for preventing people driving into floodwaters [13,17,21,23], however for child drowning, the responsibility lies with the parents or caregivers (as drivers of the vehicle) and therefore such strategies must influence their decision making. Use of a health protection model [43] may be valuable when considering appropriate strategies; as similar approaches to discouraging the behavior of driving into floodwaters have appealed to drivers to consider the risk to their own life, that of their passengers and rescuers [21].

Findings of this research suggest highlighting the risks in the context of drivers losing their own child or risking other children's lives.

Primary preventative efforts must focus around ensuring people do not drive into floodwaters [44]. However, should vehicles accidentally end up in water, if children and adolescents are old enough, they should be educated about removing their seatbelts, as one part of a strategy to enable quicker escape from the vehicle. Adult passengers should release younger children from child restraints prior to attempting to exit the vehicle [45].

*4.3. Risk of Swimming in Floodwaters*

Those in the older age groups (10–19 years) were most likely to drown in floodwaters while swimming or recreating (91% of all cases). This highlights the challenge of addressing behavior change, particularly among a group that is prone to risk taking [46]. Supervision of this group is challenging, as adolescents may be more likely to be supervised by their peers, as opposed to parents or adult caregivers. As such, there is a need to highlight flood-related drowning risk to children and adolescents in this age group in greater detail. Further research is required to explore optimal means of achieving impact on flood-related behavior among this age group. As outlined in the Hyogo Framework for Action, disaster risk reduction knowledge should be included in relevant school curricula, among use of other formal and informal channels, to educate children and young people [47].

*4.4. Social Determinants Impacting Flood Drowning Risk*

Social determinants of health, namely geographical isolation and residing in a low socio-economic area, were found to impact flood drowning risk for children and adolescents. This is consistent with research that finds increased injury risk in isolated and disadvantaged areas [48,49]. Increased risk of vehicle-related drowning during times of flood in such areas is often due to reduced infrastructure, such as bridges [50] and long travel times, making alternate routes unappealing [16]. For those of low socio-economic backgrounds, cheaper land and housing options may often be in areas that are flood-prone [51]. This, in turn, makes it is harder to get insurance for flood damage [52]. Given drowning costs the Australian economy $1.24 billion annually [53], increased investment in drowning prevention strategies—including those to mitigate the impacts of flood on adults, children, and adolescents alike—are required. Though challenging, such mitigation strategies must be upstream in their approach and target the root cause of this risk being social inequality.

*4.5. Implications for Prevention and Opportunities for Further Research*

Implications for prevention include the need to target specific ages at higher risk of flood-related drowning, as well as parents and caregivers, particularly around the recreational element to flood risk among older children and adolescents. Timing is likely to be crucial with the wet season months of November to April, particularly in the states and territories in the north of Australia which are prone to tropical rainfall, representing an opportune time to trigger public awareness messages about flood-related drowning risk.

Strategies to reduce flood-related drowning risk must address the social determinants of health, such as socio-economic disadvantage and geographical remoteness. Such strategies may include ensuring low-cost housing is not built in flood-prone areas and increased investment in infrastructure and flood management in rural and remote areas. Any such interventions must be evaluated to understand what is most effective in reducing flood-related drowning for those most at risk.

Those tasked with prevention of flood-related drowning deaths should consider recommendations already delivered for reducing such deaths, such as coronial recommendations and the recommendations of commissions of inquiry convened to examine major flood fatality events. These recommendations, such as those of the Queensland Floods Commission of Inquiry, which was handed down after a mass flash flooding event claimed the lives of 33 people in the state of Queensland in

2011 [54], provide an opportunity to link the drowning prevention sector with policy makers in other sectors where flood risk, and ultimately prevention, intersect.

Further work is also needed around exploring the types of strategies that are likely to be effective in child and adolescent flood-related drowning prevention. Typical drowning prevention strategies often recommended, such as signage, are unlikely to be effective for children and adolescents [30]. Involving children and adolescents in the development of flood mitigation and prevention strategies should be considered [41]. However, the challenge remains about how to prepare children and adolescents for a flood or disaster event without increasing complacency and potentially normalizing risky behavior around floods which may in turn increase risk of drowning.

*4.6. Strengths and Limitations*

This study is a total population survey based on detailed medico-legal investigations. It is longitudinal in nature, spanning a 16-year study period. This study includes all unintentional fatal drowning, not only the narrow definition of drowning using International Classification of Diseases (ICD) codes (W65–74 only) [8] which is vitally important for flood-related drowning deaths (X37 and X38). There are however limitations associated with this study. At the time of analysis 13.6% of cases remain open and therefore under coronial investigation. Cases which remain open at time of analysis, do not have autopsy, toxicology, and police reports available, limiting available data and any available details may change, pending the outcome of these coronial investigations. The fact that the drowning is flood-related will not change for those cases which are open. It may be that not all flood-related drowning cases among children during the study period have been identified. Using only one country's data to explore this issue limits applicability and therefore international collaboration on this topic would be of value [55]. While outside the scope of this study, it may be that over the 16-year period of data analyzed, preventative measures to address flood-related drowning may have been recommended by a coroner or implemented. This may be a worthwhile topic for a future study.

## 5. Conclusions

Flood-related drowning deaths among children and adolescents are rare yet regularly occurring. This study identified causal factors by age groups for children and adolescents impacting flood-related drowning risk that should be considered when developing prevention interventions or undertaking advocacy. These included the importance of supervision for young children, avoidance of floodwaters when driving or when recreating, and the social determinants of geographical remoteness and socio-economic disadvantage. It is hoped that, by publishing these results, it will enhance the epidemiological evidence base, as well as encourage others to publish their data on this issue, thus resulting in more effective prevention and reduction in loss of life.

**Author Contributions:** A.E.P. and R.C.F. conceptualized the study. A.E.P. cleaned and coded the cases and ran the analysis. R.C.F. assisted with analysis. Both A.E.P. and RCF prepared the original draft manuscript. A.E.P. prepared the data visualization. A.E.P and R.C.F provided critical revision of the manuscript. All authors approve the submitted manuscript.

**Funding:** This research received no external funding.

**Acknowledgments:** This research is supported by Royal Life Saving Society – Australia to aid in the reduction of drowning. Research at Royal Life Saving Society – Australia is supported by the Australian Government.

**Conflicts of Interest:** The authors declare no conflict of interest.

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
