# Peer review of "Exploring Flood-Related Unintentional Fatal Drowning of Children and Adolescents Aged 0–19 Years in Australia"

_safety, 2019_

Round 1
Reviewer 1 Report
Well written.
Author Response
5-07-2019
Thank you to reviewer 1 for their comments. We note there are no suggested revisions made by reviewer 1. We have however done a spell check to ensure no errors in the final submission. We thank the reviewer for their feedback and support of the article.
Reviewer 2 Report
in attachment.

Reviewer 3 Report
Thank you for providing me with the opportunity to review this paper. Greater attention to grammar is required throughout the manuscript. Further, a greater appreciation of the role of the social determinants of health in prevention is required for this paper to make a more insightful contribution to the literature. Below, I specify the changes I deem necessary prior to publication.
p. 1 The abstract makes no mention of Australia – this must be included.
Why aren’t flood-related drownings included in WHO estimates? Explain
“impacting” should be “affected”
p. 2 “This data” should be “These data” or “This datum”
p. 3 Why did you choose a 16 year period?
p. 3 “Across the 16 year study period, a total of 44 children and adolescents aged 0-19 years drowned
due to flooding, this represent[s]…” replace second comma with a semi-colon and add an “s” to represents
p. 5 “Rivers, creeks and streams were the leading location” – shouldn’t it be leading locationS?
p. 7 Change “impacted” to “affected”
p. 7 What is an MFE?
p. 8 “this data” should be “these data”
p. 8 i.e. is usually followed by a comma (i.e.,)
p. 8 Flood-related drowning among young children is often as a result of unintentional entry into floodwaters. Consider revising to “…often results from unintentional…”
p. 8 “Active adult supervision is a vital part of a multi-faceted drowning prevention strategy for young children.” Please provides references.
p. 8 “For natural bodies of water, fencing (restricting access) of the water body is often not practical. As such a child safe play area (i.e. a barrier around the child) is a better strategy.” This seems to assume that the child is playing prior to drowning. This is a problematic assumption, especially given that you state that activities prior to drowning are poorly understood. I don’t think this sentence adds much. I would delete it.
p. 8 “Finding of this research” should be FindingS
p. 8 “However should…” requires a comma after However
p. 8 Is releasing the seat belt enough? Escape from a sinking vehicle requires other skills and knowledge that can also be taught.
p. 9 particularly in a group that are prone to risk taking – group is singular, so it has to be IS and not ARE
p. 9 “As such” needs to be followed by a comma (same thing earlier in the manuscript)
p. 9 “Floodwaters, particularly in Australia, are an irregular occurrence and represent a fascination to children and adolescents that makes them attractive.” Please provide references for both assertions in this sentence.
p. 9 “It is important that parents and caregivers instill in children and adolescents, the risks of entering floodwaters.” Delete comma
p.9 “Social determinants of health, namely geographical isolation and residing in a low socio- economic area[,] were found to impact flood drowning risk for children and adolescents.” See addition of comma
p. 9 “Increased risk of drowning during times of flood in such areas is often due to reduced infrastructure, such as bridges [49] and long travel times, making alternate routes unappealing [16].”
-can you more specifically tie this statement to child drowning?
p. 9What does flood insurance have to do with your argument? I’m confused as to how it is relevant. Shouldn’t the argument not really be about flood mitigation by about how reducing flood-related drowning means addressing the root cause of the problem: low access to the social determinants of health? You need to go further upstream in your thinking.
p. 9 The “fascination with floodwaters” argument seems very, very weak to this reader. Provide evidence to delete.
p. 9 Your implications for prevention again don’t include addressing the SDHs, which are what put people at great risk. Please address this.
p. 9 Why is complacency a problem? Explain.
Reviewer 4 Report
Title: please change to children and young adults aged 0-19 years and add the region you investigated
Abstract: explain the region you investigated
Line 115 and following should be a separate section 'Statistical Analyses'
Line 124 and following should be put at the beginning of the method section
Line 211 start the discussion with the main findings and make then sections discussing each major finding
Round 2
Reviewer 2 Report
The authors were able to respond to most of the remarks and questions made.
Author Response
23-07-2019
Thank you to reviewer 2 for their comments on our revision. We note there are no further suggested revisions made by reviewer 2. We thank the reviewer for their feedback and support of the article.
Reviewer 3 Report
The assertions that the authors make about children being fascinated by floods remains in the paper, despite the fact that they recognize that they have no data to support this claim and that it is merely their assertion. This is dangerous. The inclusion of "may" does not undo the problematic notion of this assertion, for which they have no data. This needs to be removed. It totally undermines an otherwise solid paper. If they want to make that claim, interview kids and parents and get some data to support it; otherwise, it is unscientific and also harms the journal's credibility.
Author Response
23-07-2019
Thank you to reviewer 3 for their comments on our revision. We note the reviewer advises us again to remove our claims regarding children and their fascination with floodwater. While we note that there is research around children’s fascination with water in general, contributing in part to increased risk of drowning in locations such as home swimming pools we acknowledge the lack of data around the authors’ hypothesis regarding children’s fascination with floodwaters, especially in a drought prone country.
As such we have removed the two sentences that mentioned this potential fascination from section 4.3 of the discussion, as well as the paragraph referring to the same in section 4.4 social determinants impacting flood drowning risk. Please see the revised version of the manuscript.
We thank the reviewer for their feedback and support of the article.
Reviewer 4 Report
no further comments
Author Response
23-07-2019
Thank you to reviewer 4 for their comments on our revision. We note there are no further suggested revisions made by reviewer 4. We thank the reviewer for their feedback and support of the article.
Round 3
Reviewer 3 Report
I thank the reviewers for their changes, as it really strengthens the paper's credibility (and thus the authors').